# Urate-Lowering Therapy Use among US Adults with Gout and the Relationship between Patients' Gout Treatment Status and Associated Comorbidities

Marcos Ortiz-Uriarte [1], Jeanlouis Betancourt-Gaztambide [1], Alexandra Perez [1] and Youssef M. Roman [2,*]

[1] Department of Sociobehavioral and Administrative Pharmacy, Nova Southeastern University College of Pharmacy, 3200 S. University Dr., Davie, FL 33328, USA

[2] Department of Pharmacotherapy and Outcomes Science, Virginia Commonwealth University School of Pharmacy, 410 N 12th Street, Richmond, VA 23298, USA

\* Correspondence: romany2@vcu.edu

**Abstract:** Gout is one of the most common inflammatory conditions with a growing global prevalence. Individuals with gout are at higher risk of developing chronic conditions, such as diabetes, chronic kidney disease (CKD), and cardiovascular diseases. In this study, the association between urate-lowering therapy (ULT) use and the prevalence of these conditions was evaluated. This observational cross-sectional pharmacoepidemiologic study used the 2013–2018 biannual cycles of the National Health and Nutrition Examination Survey. The inclusion criteria were adults that were 30 years of age or older that had a diagnosis of gout. The association between patients' ULT treatment status and dyslipidemia, coronary heart disease, heart failure, hypertension, and chronic kidney disease was evaluated as well as its association with select clinical laboratory biomarkers. The prevalence of ULT use was 28.9% (95% CI 24.3–33.9%). Those receiving ULT had a higher prevalence of CKD diagnoses, of a college graduate or higher and of health insurance coverage, and they were older obese males. There was no significant association between ULT use and the prevalence of heart failure, coronary heart disease, hypertension, or dyslipidemia ($p > 0.05$). Those receiving ULT had lower high-sensitivity c-reactive protein levels compared to those who were not on treatment (4.74 versus 7.21 mg/L, $p = 0.044$). LDL and total cholesterol were significantly lower among those receiving ULT treatment ($p < 0.05$). ULT use continues to be low among US individuals diagnosed with gout. Socioeconomic factors may influence patients' ULT treatment status. Also, gout risk factors, including obesity, male sex, and CKD, are associated with receiving ULT. While our findings may have reflected the guideline recommendations for ULT use in CKD patients, worsening kidney functions while receiving ULT is unlikely. Gout patients receiving ULT may garner added health benefits beyond lower urate levels. Further research is necessary to determine the long-term impact of ULTs on lipid fractions, kidney functions, and other cardiovascular biomarkers.

**Keywords:** gout; urate-lowering therapy; serum uric acid; pharmacoepidemiology; lipids; hs-CRP; gout comorbidities

## 1. Introduction

Gout is the most common inflammatory arthritic condition, affecting more males than females and disproportionately impacting select racial and ethnic groups [1–3]. Sustained high serum uric acid (SUA) levels, a condition known as hyperuricemia, leads to the formation of monosodium urate crystals—the hallmark of developing gout [4]. The formation and the deposition of monosodium urate crystals in and around the joints are responsible for acute gout flares. Reports on the prevalence and incidence of gout vary widely depending on the population studied and the methods employed; however, the prevalence rates range from <1% to 6.8%, and the incidence range is 0.58–2.89 per 1000 person-years [5]. The clinical symptoms of gout progress through different stages, including asymptomatic

hyperuricemia, monosodium urate crystal formation and deposition, and intermittent gout flares. Optimal gout management requires sustained urate level reduction to enhance monosodium urate crystal dissolution and to prevent recurrent gout flares [6]. For recurring gout flare prevention and treatment, the American College of Rheumatology guidelines recommend urate-lowering therapy (ULT) with xanthine oxidase inhibitors, allopurinol or febuxostat, as the first-line option for achieving a goal serum urate level of less than 6 mg/dL [7]. Probenecid is a second-line agent and can be combined with xanthine oxidase inhibitors in cases where monotherapy is not enough to reach treat-to-target urate levels. Despite the high incidence of gout in the United States population, studies suggest that most patients with gout do not receive the appropriate treatment with which to manage their disease. An observational study published in 2019 using data from the National Health and Nutrition Examination Survey using 2007–2016 survey waves showed that only one-third of those diagnosed with gout were treated with ULT [5]. Additionally, it was estimated that only 37.7% of gout patients were meeting their therapeutic serum urate goal [8]. Despite strong recommendations, the pharmacologic treatment of gout is often limited to the use of anti-inflammatory agents for symptomatic relief. This approach can provide comfort to the patient but ignores the underlying cause of the elevated serum urate level, leading to recurring flares and further joint damage. The short-term complications of gout include the development of subcutaneous tophi, a granulomatous reaction to the monosodium urate crystals deposited in the joints, and kidney stones [9–11].

Patients with gout (or chronic hyperuricemia) also have a higher risk of developing cardiovascular diseases and are at an increased risk of major adverse cardiovascular events (acute coronary syndrome, stroke, arrhythmia, or peripheral artery disease) [12–17]. Furthermore, gout is an independent risk factor for developing type 2 diabetes or hypertension [16–20]. A limited number of studies have evaluated the impact of ULT and the long-term clinical outcomes among patients with gout. Allopurinol and febuxostat have been associated with improved treatment outcomes in heart failure patients and a reduced risk of acute cardiovascular events [21,22]. The American College of Rheumatology conditionally recommends ULT for those experiencing their first gout flare and for those with CKD ≥ 3 and SU > 9 mg/dL or urolithiasis. While a recent meta-analysis found that the use of allopurinol was associated with changes in patients' blood glucose parameters [23], the association between ULT use and clinical biomarkers, such as hemoglobin A1C, C-reactive protein (CRP), and lipid traits, remains ill defined. Evaluating these clinical outcomes is critical for improving adherence to ULT and for providing real-world evidence for optimal gout management, especially in the context of multiple comorbidities [6]. Despite the aforementioned observations, the relationship between patients' gout treatment status and the prevalence of chronic conditions, such as ischemic heart disease and heart failure, remains controversial, and no recent studies have explored the association between the prevalence of these conditions and patients' gout treatment status. To address these gaps, we evaluated the relationship between the prevalence of coronary heart disease, heart failure, hyperlipidemia, hypertension, chronic kidney disease, and clinical biomarkers and patients' gout treatment status among US adults with a diagnosis of gout.

## 2. Materials and Methods

This observational cohort study used data from the National Health and Nutrition Examination Survey (NHANES). NHANES is a cross-sectional prospective continuous health survey that aims to assess the health and nutrition of the noninstitutionalized civilian population in the United States. NHANES conducts at-home health interviews and physical examinations at mobile centers. Questionnaire, laboratory, and physical examination data are available in 2-year survey waves. Data from the NHANES survey waves of 2013–2014, 2015–2016, and 2017–2018 were combined in this study. We included adults who were 30 years or older and who had answered 'yes' when being asked whether a doctor had told him/her that they had gout. The participant selection process is summarized in Figure 1.

All data used in the study is publicly available and deidentified, and no institutional review board approval was required (45 CFR §46.102(f)).

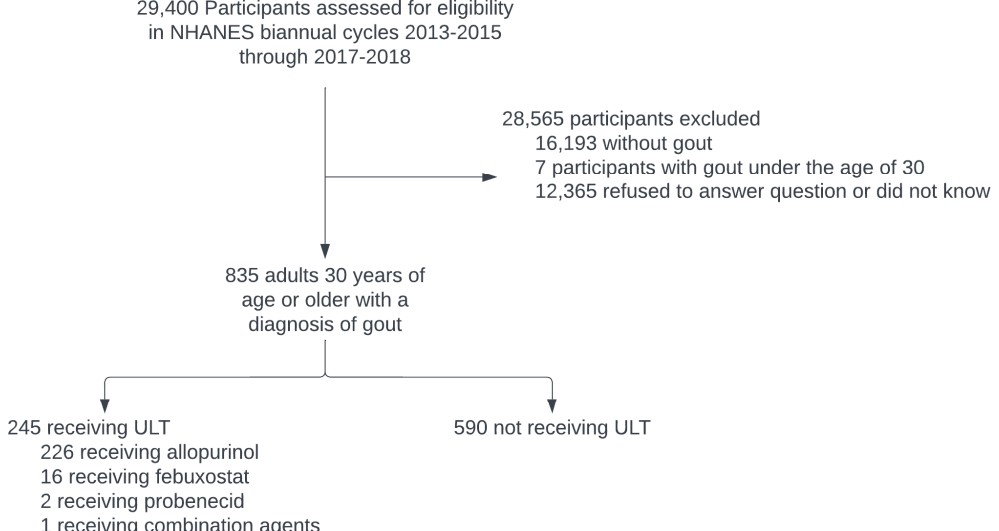

**Figure 1.** Sample selection process and ULT use breakdown. ULT represents urate-lowering therapy, and NHANES represents National Health and Nutrition Examination Survey.

### 2.1. Assessment of Urate-Lowering Therapy Use

Urate-lowering therapy (ULT) was defined as the use of any ULT medication (yes/no), including allopurinol, febuxostat, probenecid, or combination agents. Combination products were lesinurad/allopurinol and colchicine/probenecid. ULT use data was obtained from the medication file of each 2-year cycle, which collected information on bottle-verified use of prescribed medications within the 30 days before the survey interview.

### 2.2. Evaluation of Chronic Conditions and Clinical Biomarkers

Participants with comorbid dyslipidemia, coronary heart disease, heart failure, hypertension, or chronic kidney disease were identified through individual health survey questions asking whether a doctor had ever told them that they had any of those conditions. Biomarkers of interest were serum uric acid, C-reactive protein (CRP), glycohemoglobin, high-density lipoprotein, low-density lipoprotein, and total cholesterol levels, which were obtained from the laboratory component. C-reactive protein levels were only available in the 2015–2016 and 2017–2018 survey waves. NHANES uses standardized methods to collect blood and urine samples. Other biomarkers included eGFR and systolic and diastolic blood pressures. eGFR was calculated using the Modification of Diet in Renal Disease (MDRD) equation and the respective demographic and laboratory data. Systolic and diastolic blood pressure levels were obtained from the physical examination, and the average of two different readings for each participant were used in this study.

### 2.3. Statistical Analyses

Prevalence of ULT use overall and among participants with the mentioned chronic conditions was estimated and reported using an unweighted sample count and nationally weighted percentage. Count and national percent were also used to describe the use of ULT by individual agents, prevalence of comorbid chronic conditions, and the proportion of participants meeting the uric acid treatment goal (serum uric acid <6 mg/dL or ≥6 mg/dL) per gout treatment status. Mean and standard error (SE) were used to report all continuous laboratory marker values per gout treatment status. Chi-square tests were used to evaluate the association between ULT use and biannual survey years and to compare sociodemographic factors (sex (male/female), race/ethnicity (Hispanic, non-Hispanic White, non-Hispanic Black, non-Hispanic Asian, or other race), education (high school or less,

some college, or college graduate or higher), health insurance coverage (yes/no), and body mass index (<25 kg/m$^2$ or ≥25 kg/m$^2$)) with patients' gout treatment status. Univariable and multivariable logistic regression models were developed to evaluate the association between ULT use and the prevalence of comorbid conditions. Multivariable models used age, sex, race/ethnicity, diagnosis of type 2 diabetes, health insurance status, weight category, level of education, and duration of gout as covariates. Univariable and multivariable linear regression models were used to calculate the beta coefficients of biomarkers and their respective 95% confidence intervals per ULT status. The adjusted models used the same covariables as in the multivariable logistic regression model. Patients diagnosed with diabetes may garner added benefits due to the use of various medications that may influence the outcomes of interest. Additionally, diabetes diagnoses may introduce bias by virtue of routine care, and, therefore, adjusting for diabetes was warranted. Crude and adjusted odds ratios and their respective 95% confidence intervals were reported. Odds ratios represented the odds of having the chronic condition of interest per gout treatment status. Beta coefficients represented the degree of change per ULT status for each biomarker. An alpha value of 5% was used to determine statistical significance. All statistical tests were adjusted for complex sampling design, and estimates were nationally representative. IBM SPSS version 27 was used to run all statistical analyses.

## 3. Results

### 3.1. Sample Characteristics and ULT Use

The study cohort included 835 adults who were 30 years or older and who had been told by a doctor that they had gout. The sample characteristics are shown in Table 1. Individuals receiving ULT had a higher mean age (64.86 ± 1.05 versus 60.77 ± 0.62, $p < 0.001$) and were more likely to be male (75.8% versus 63.8%, $p = 0.023$). There was no difference between racial/ethnic groups ($p > 0.05$). Individuals receiving ULT were more likely to have a higher level of education, health insurance coverage, and a body mass index (BMI) of 25 kg/m$^2$ or greater ($p < 0.05$) (Table 1). The mean duration of gout was 13.4 years, and it did not significantly differ between the groups ($p > 0.05$). Among the 835 participants in our cohort, 245 received ULT (28.9%, 95% CI 24.3–33.9). The use of ULT did not change significantly over time from 2013 to 2018 ($p > 0.05$, data not shown). The most reported ULT medication was allopurinol (91.4%, 95% CI 84.9–95.3) followed by febuxostat (7.3%, 95% CI 3.5–14.8), probenecid (1%, 95% CI 0.3–3.7), and combinations drugs (0.2%, 95% CI 0–1.7). There was a significant difference in the urate levels between the ULT treatment statuses, the levels being lower for those receiving treatment (5.81 mg/dL versus 6.57 mg/dL, $p < 0.001$). Of those receiving ULT, 63.9% met their serum uric acid level goal compared to 39.1% who reached their goal without the use of ULT ($p < 0.001$).

**Table 1.** Sample characteristics of U.S. adults who were 30 years or older and who were told by a physician that they had gout.

| Variable | Receiving ULT | | *p*-Value * |
| :---: | :---: | :---: | :---: |
| | **Yes** <br> **N = 245** | **No** <br> **N = 590** | |
| Age (mean, SE) | 65, 1.05 | 61, 0.62 | <0.001 |
| Sex (n, %) <br> Male <br> Female | <br> 180, 75.8% <br> 65, 24.2% | <br> 385, 63.8% <br> 205, 36.2% | <br> 0.023 |
| Race/Ethnicity (n, %) <br> Hispanic <br> Non-Hispanic White <br> Non-Hispanic Black <br> Non-Hispanic Asian <br> Other races | <br> 29, 6.9% <br> 102, 69.8% <br> 65, 11.9% <br> 37, 7.3% <br> 12, 4% | <br> 100, 8.7% <br> 248, 70.1% <br> 161, 13.1% <br> 56, 4.3% <br> 25, 3.9% | <br> <br> 0.454 |

**Table 1.** *Cont.*

| Variable | Receiving ULT | | *p*-Value * |
| | Yes N = 245 | No N = 590 | |
|---|---|---|---|
| Highest Level of Education (n, %) | | | |
| High school or less | 117, 37.4% | 282, 39.4% | |
| Some college | 76, 26.4% | 187, 37.1% | 0.048 |
| College graduate or higher | 52, 36.1% | 121, 23.5% | |
| Covered by Health Insurance (n, %) | | | |
| Yes | 242, 99.6% | 528, 88.4% | < 0.001 |
| No | 3, 0.4% | 62, 11.6% | |
| Weight Status (n, %) | | | |
| BMI $\leq 24.9$ kg/m$^2$ | 27, 7.0% | 102, 16.4% | 0.010 |
| BMI $\geq 25$ kg/m$^2$ | 200, 93.0% | 454, 83.6% | |
| Duration of Gout (years) (mean, SE) | 13.5, 1.10 | 14.6, 0.80 | 0.413 |
| Serum Uric Acid Target (<6 mg/dL) (n, %) | | | |
| Yes | 133, 63.1% | 201, 39.1% | <0.001 |
| No | 92, 36.9% | 334, 60.9% | |
| Serum Urate (mg/dL) (mean, SE) | 5.81, 0.11 | 6.57, 0.10 | <0.001 |

ULT represents urate-lowering therapy, SE represents standard error, and BMI represents body mass index. * Chi-squared tests were used to evaluate the relationship between ULT use and categorical variables. Independent *t*-tests were used to evaluate the relationship between ULT use and continuous variables. *p* values < 0.05 were considered statistically significant.

### 3.2. Prevalence of Chronic Comorbid Conditions per Gout Treatment Status

Among those who met the inclusion criteria, 40.8%, 14.6%, 14.9%, 71.9%, and 15.0% reported having been diagnosed with dyslipidemia, coronary heart disease, heart failure, hypertension, and chronic kidney disease, respectively. There was no significant association between ULT use and dyslipidemia, coronary heart disease, heart failure, and hypertension in the univariable model or the multivariable model (*p* > 0.05). There was a significant association between chronic kidney disease (CKD) and ULT treatment status (OR of 2.38, 95% CI 1.32–4.30). The association remained significant after adjusting for age, race/ethnicity, sex, BMI, level of education, health insurance status, duration of gout, and diagnosis of type 2 diabetes (OR of 2.35, 95% CI 1.07–4.31, Table 2).

**Table 2.** Odds ratios for the prevalence of comorbid conditions among adults receiving urate-lowering therapy who were 30 years or older and who had been told by a doctor that they had gout.

| Variable | Receiving ULT | | Crude OR (95% CI) | Adjusted OR (95% CI) |
| | Yes N = 245 | No N = 590 | | |
|---|---|---|---|---|
| Type 2 Diabetes Diagnosis | | | | |
| Yes (n, %) | 69, 21.8% | 152, 18.5% | 1.22 (0.684–2.19) | N/A |
| No (n, %) | 176, 78.2% | 438, 81.5% | | |
| Heart Failure Diagnosis | | | | |
| Yes (n, %) | 44, 13.7% | 80, 9.2% | 1.56 (0.834–2.93) | 1.33 (0.74–2.40) |
| No (n, %) | 198, 86.3% | 507, 90.8% | | |
| Coronary Heart Disease Diagnosis | | | | |
| Yes (n, %) | 46, 16.9% | 76, 14.0% | 1.25 (0.69–2.26) | 0.95 (0.48–1.89) |
| No (n, %) | 197, 83.1% | 511, 86.0% | | |
| High Cholesterol Diagnosis | | | | |
| Yes (n, %) | 151, 61.1% | 338, 58.9% | 1.09 (0.66–1.80) | 0.84 (0.51–1.40) |
| No (n, %) | 90, 38.9% | 251, 41.1% | | |

**Table 2.** *Cont.*

| Variable | Receiving ULT | | Crude OR (95% CI) | Adjusted OR (95% CI) |
| --- | --- | --- | --- | --- |
| | Yes N = 245 | No N = 590 | | |
| High Blood Pressure Diagnosis | | | 1.50 (0.811–2.78) | 1.20 (0.63–2.30) |
| Yes (n, %) | 199, 74.0% | 401, 65.5% | | |
| No (n, %) | 46, 26.0% | 189, 34.5% | | |
| Chronic Kidney Disease Diagnosis | | | 2.38 (1.32–4.30) | 2.35 (1.07–4.31) |
| Yes (n, %) | 56, 19.2% | 71, 9.0% | | |
| No (n, %) | 190, 80.8% | 519, 91.0% | | |

ULT represents urate-lowering therapy, OR represents odds ratio, and CI represents confidence interval. Adjusted logistic regression models used age, sex, race/ethnicity, BMI, level of education, health insurance coverage, duration of gout, and type 2 diabetes diagnosis as covariates.

### 3.3. Evaluation of Clinical Biomarkers per Gout Treatment Status

Total and LDL-cholesterol levels were significantly higher among those who were not receiving ULT, but no difference was observed across triglyceride and HDL levels. In the adjusted model, those receiving ULT also had significantly lower high-sensitivity CRP levels of 4.74 mg/L compared to 7.21 mg/L for those not receiving ULT ($p = 0.044$). There were no statistically significant differences in glycohemoglobin, eGFR, or systolic or diastolic blood pressures between the gout treatment statuses in the adjusted model (Table 3).

**Table 3.** Clinical biomarkers among adults who were 30 years or older who had been told by a doctor they had gout.

| Variable | Receiving ULT | | Unadjusted Model | | Adjusted Model | |
| --- | --- | --- | --- | --- | --- | --- |
| | Yes (n = 245) (Mean, SE) | No (n = 590) (Mean, SE) | Beta Coefficient (95% CI) | *p*-Value * | Beta Coefficient (95% CI) | *p*-Value * |
| C-Reactive Protein (mg/L) | 4.74, 0.64 | 7.21, 0.98 | 2.59 (−1.75–6.92) | 0.223 | 2.46 (0.08–4.85) | 0.044 |
| Glycohemoglobin (%) | 6.21, 0.05 | 6.29, 0.09 | −0.70 (−0.35–0.21) | 0.620 | 0.02 (−0.22–0.25) | 0.872 |
| HDL-Cholesterol (mg/dL) | 46.23, 1.82 | 49.15, 0.96 | 3.88 (0.51–7.25) | 0.025 | 2.92 (−1.24–7.07) | 0.164 |
| LDL-Cholesterol (mg/dL) | 96.90, 3.39 | 108.86, 3.45 | 16.05 (6.12–25.98) | 0.002 | 11.96 (1.08–22.84) | 0.032 |
| Triglycerides (mg/dL) | 161.75, 23.32 | 144.50, 7.20 | −22.94 (−64.54–18.66) | 0.272 | −17.25 (−66.54–32.05) | 0.484 |
| Total Cholesterol (mg/dL) | 177.94, 3.78 | 188.88, 2.95 | 16.41 (7.55–25.27) | <0.001 | 10.95 (0.635–21.26) | 0.038 |
| Systolic Blood Pressure (mmHg) | 131.57, 2.10 | 131.44, 1.26 | −0.64 (−5.71–4.44) | 0.802 | −0.12 (−5.53–5.29) | 0.964 |
| Diastolic Blood Pressure (mmHg) | 71.07, 1.38 | 72.15, 0.91 | 2.87 (−0.31–5.76) | 0.052 | 1.09 (−2.09–0.69) | 0.494 |
| eGFR (mL/min/1.73 m$^2$) | 71.31, 1.83 | 74.37, 1.23 | 6.70 (2.61–1.44) | 0.014 | 3.05 (−1.82–7.93) | 0.214 |

ULT represents urate-lowering therapy, SE represents standard error, HDL represents high-density lipoprotein, LDL represents low-density lipoprotein, eGFR represents estimated glomerular filtration rate. * Linear regression models were used to evaluate the relationship between ULT use (independent variable) and clinical biomarkers (dependent variable). Adjusted models used age, sex, race/ethnicity, BMI, level of education, health insurance coverage, duration of gout, and type 2 diabetes diagnosis as covariates. Beta coefficients and 95% CIs were calculated by using the univariable and multivariable linear regression models. Means and SEs were calculated by using descriptive statistics from the multivariable linear regression models. *p* values < 0.05 were considered statistically significant.

## 4. Discussion

This study was the first pharmacoepidemiologic study to assess the prevalence of chronic comorbid conditions per gout treatment status among US adults. In this nationally representative sample of US adults with gout, the prevalence of ULT use was 28.9%. This estimate fell into the lower end of what has been previously reported in published epidemiologic studies [5]. The relatively low prevalence of ULT use among adults with gout and the difference in serum uric acid levels between those receiving ULT and those not receiving ULT suggested that gout remains suboptimally managed in a sizable proportion of patients. Poor gout management translates into a higher risk of frequent gout flares and the development of gout-related complications, such as nephrolithiasis, hypertension, and cardiovascular events [22]. Augmented gout care could improve gout management and could increase the uptake of ULT among patients with gout. Different care models involving nurses and pharmacists have shown a great potential for increasing the proportion of patients receiving ULT and for achieving SU level targets [24]. Efforts focused on patients' understanding of the disease and adherence to ULT are needed to minimize the burden of suboptimal gout management. Also, gout is primarily managed in the primary care setting, and, therefore, educating primary providers on optimal gout management strategies may improve gout treatment outcomes and may reduce gout flare recurrence.

Consistent with previous reports, our patients with gout who received ULT had distinct characteristics compared with those who did not [25]. In our study, more obese older male patients received ULT than those who did not. An older age, male sex, and higher BMI are associated with an increased risk of developing gout. Knowledge of gout risk factors can prompt healthcare providers to initiate ULT in a timely manner among high-risk population groups. Moreover, health insurance coverage and a college degree or higher were significantly more prevalent among those who received ULT compared with those who did not. This highlights the role of socioeconomic factors and the social determinants of health in achieving equitable access to care. Moreover, among patients receiving ULT, approximately one-third of them did not achieve their uric acid target. Several factors have been attributed to a large portion of gout patients not achieving their uric acid therapeutic target. For example, a low adherence to ULT, a lack of frequent urate level monitoring, physician inertia with respect to escalating ULT dosing, and genetic variability in response to ULT have been previously reported [6,26–28].

This study found no association between the chronic comorbidities associated with gout and patients' ULT treatment status except for its association with chronic kidney disease. Those who received ULT were more likely to have been diagnosed with CKD compared to those who did not receive ULT. Prompt treatment with ULT in individuals with impaired kidney functions, especially in those with moderate-to-severe CKD, is conditionally recommended by the most recent American College of Rheumatology gout management guidelines and may aid in preventing frequent gout flares or worsening kidney functions [7]. Although CKD is common among gout patients, especially late in the course of the disease, there is limited data regarding the renal effects of allopurinol on gout patients with normal renal functions. A large prospective propensity-score-matched study of individuals diagnosed with gout who initiated allopurinol ($\geq$300 mg/day) compared with those who did not was conducted in the United Kingdom. A follow-up on the use of allopurinol at a dose of at least 300 mg/day over five years was associated with a lower risk of developing stage 3 CKD or higher compared with nonusers with a hazard ratio of 0.87 (95% CI 0.77–0.97) [29]. Therefore, our study observations could be the result of the underlying recommendations in most guidelines that ULT is recommended for gout patients with CKD. Nonetheless, allopurinol's active metabolite, oxypurinol, is renally eliminated. Thus, declining kidney functions can result in the accumulation of the active metabolite in the kidney, which may increase the risk of nephrotoxicity. While the nephrotoxic effect of allopurinol is rare and yet to be determined, there is growing evidence to suggest that ULT may preserve kidney functions [30]. This might explain the higher use of ULT among patients with impaired kidney functions. Despite the relationship

between gout and chronic diseases reported in the literature [14,17], there was no significant association between ULT use and heart failure, coronary heart disease, hypertension, and dyslipidemia comorbidities in this study.

The association between gout and cardiovascular disease (CVD) remains too controversial to recommend frequent SUA monitoring to follow an individual's risk of CVD or to recommend using ULT for hypertension. Although SUA can be an independent risk factor for developing CVD or kidney disease, SUA may not be the direct cause of kidney disease. However, SUA concentrations are believed to lead to hypertension, which may more significantly affect the kidney than SUA itself. Therefore, the causal relationship between SUA and CVD requires a reappraisal and further studies to show the direct effect of SUA on CVD and kidney disease [31]. A randomized crossover-controlled trial of relatively healthy young adults (18–40 years) without chronic kidney disease showed that allopurinol therapy did not affect systolic or diastolic blood pressures [32]. However, the study found a significant effect of allopurinol treatment on flow-mediated dilation, supporting the hypothesis that SUA affects endothelial functions [32]. Consistent with the previous report, our results showed that receiving ULT treatment was not associated with lower diastolic or systolic blood pressures compared with those not receiving ULT. However, improved clinical outcomes in patients with cardiovascular disease receiving ULT have been reported in the published literature [21,22]. A case-controlled study of heart failure patients with comorbid gout showed that the use of allopurinol was associated with a reduced rate of readmissions and all-cause death [21]. Febuxostat has also shown a similar effect on cardiovascular outcomes in the setting of hyperuricemia with or without gout [33,34]. Nevertheless, while decreased markers of oxidative stress have been demonstrated using ULT, these changes have not uniformly correlated with structural or functional improvements among patients with HF. More work is needed to understand the complex pathophysiology underlying increased xanthine oxidase activity and the role of ULT in ameliorating oxidative stress [35].

High serum urate levels have been reported to predict the development of multiple cardiometabolic disorders via inflammation [36]. Though the exact mechanism remains elusive, high SUA levels can damage smooth muscle cells, causing the release of high-sensitivity C-reactive protein (hs-CRP) and monocyte chemoattractant protein-1, both of which have a major role in initiating atherosclerotic lesions [36,37]. The association between hyperuricemia and increased inflammation allowed for the investigation of the effect of ULT on the major inflammatory cytokines [38]. In line with previous reports, our study found that individuals receiving ULT had lower hs-CRP levels compared to those not receiving ULT [38]. Elevated levels of CRP, an inflammatory biomarker, have been associated with gout and hyperuricemia [39]. Moreover, a reduction in CRP levels has been positively associated with an improvement in endothelial functions and blood pressure control [40,41]. Xanthine oxidase inhibitors can reduce inflammation via the regulation of lipolysis by the adipocytes and, thus, may be beneficial in the prevention of cardiovascular or metabolic diseases [42]. Our study showed that the treatment of gout with ULT did produce significant reductions in hs-CRP and suggested that high uric acid levels may lead to sterile inflammation. However, further studies are needed to evaluate whether these reductions could be associated with improved outcomes in patients with comorbid chronic conditions.

The association between gout and dyslipidemia has been previously reported [43]. A case-control study of Korean participants showed that patients with gout had 1.43 OR (95% CI 1.37–1.49) of having dyslipidemia in a fully adjusted model. Nevertheless, the exact mechanism by which elevated serum urate can modulate LDL-C and TG levels is unclear. Furthermore, limited studies have investigated the effect of ULT on lipid fractions. In our study, we explored the impact of ULT use on select clinical lipid biomarkers, and, despite the lack of an observable association between ULT use and the diagnosis of dyslipidemia in our analysis, those receiving ULT had significantly lower LDL and total cholesterol levels when compared to those not receiving ULT. The lower lipid levels among those

receiving ULT suggested that gout patients with comorbid dyslipidemia may see an added benefit from using ULT. The association between ULT use and lower lipid levels may have been confounded due to the dietary habits among the individuals diagnosed with gout or excess comorbidities requiring additional therapies. Nevertheless, our findings were in line with previous reports suggesting that ULT was associated with a significant reduction in cholesterol and triglyceride levels at 3–5 weeks in Chinese patients, even when lipid-lowering therapy (LLT) was required [44]. Particularly, febuxostat was the only ULT that reduced both the total cholesterol and triglyceride levels in the absence of LLT [44].

Another large retrospective study identified that Taiwanese patients without antigout treatments had greater than a three-fold higher risk of hyperlipidemia compared with patients without gout [45]. Patients receiving ULT, however, had a significantly lower risk of hyperlipidemia than gout patients without ULT (aHR < 0.90) [16]. Furthermore, in vitro studies have shown that receiving antigout treatment decreases the expression of the hepatic genes related to lipogenesis in hepatic cells, indicating that gout patients receiving ULT could have a lower risk for developing hyperlipidemia [45]. Specifically, using the HepaRG cell line treated with antigout therapy for 24 h, antigout drugs significantly reduced the expression of lipogenic-related genes, including *LXRα, SREBP-1c, SCD, FAS, FAE, ACLY*, and *ACC*, compared with the control [45]. The reduced expression of lipogenic-related genes may lead to an improved blood lipid profile with ULT. Another potential mechanism to explain these results is the genetic intersection between the risk of developing gout and the response to LLT. To explain this, the genetic polymorphism in *ABCG2* (rs2231142C>T) is associated with a 3–4-fold higher risk of developing gout and a progression from hyperuricemia to gout [46–48]. The same genetic polymorphism has been implicated in a greater reduction in LDL-C levels among patients receiving statins, especially rosuvastatin [49–51]. Collectively, the combined effects of reduced lipogenesis, because of antigout treatment, and the possible genetic predisposition for gout may render patients with gout to garner greater benefits from LLT. Nonetheless, it is important to bear in mind that we examined the association between ULT and serum lipid changes on a cross-sectional basis; therefore, well-designed prospective studies are needed to systematically evaluate the effect of ULT use on dyslipidemia.

*Limitations*

Our analyses had some limitations. First, due to the cross-sectional nature of our data, the cause–effect relationships between patients' gout treatment status and the studied conditions and clinical laboratory markers could not be established. Additionally, data on gout severity (e.g., presence of subcutaneous tophi, frequency of gout flares, and other data) were not gathered by the NHANES. Patients' history of chronic conditions relied on self-reports, so some misclassification errors and recall biases may have occurred. Unmeasured parameters, such as the use of antihyperlipidemic, antidiabetics, or antihypertensives, were not evaluated and may have confounded the relationships studied here. This study focused on the use of urate-lowering therapy only; therefore, the use of on-demand pain medications or anti-inflammatory agents such as steroids was not evaluated. However, our study attempted to reduce the confounding effects of medications by adjusting for multiple comorbidities and other demographics that may have influenced patients' access to medications. Despite being representative of the US population, our study sample remained limited to robustly estimate the association between patients' ULT treatment status and other comorbidities. Finally, the NHANES does not collect data on medication adherence, length of therapy, or medication doses; thus, the assessment of the optimal management strategies for patients being treated with ULT could not be evaluated in this study. Despite the limitations described, this study produced nationally representative estimates and utilized multivariable logistic regression models to adjust for several confounding factors. Data on clinical laboratory markers were obtained from objective laboratory measurements, and the results were also generalizable to the ambulatory US population.

## 5. Conclusions

This is the first pharmacoepidemiologic study to evaluate the association between patients' gout treatment status and the major clinical cardiometabolic biomarkers and cardiovascular–renal comorbidities among US adults with gout. The low prevalence of ULT use could partly explain the higher mean serum urate levels among those not receiving ULT, suggesting that gout remains suboptimally managed in a large proportion of patients. A higher prevalence of ULT use among those with a diagnosis of CKD is likely to be driven by the recent data suggesting that ULT use may improve outcomes and help preserve kidney functions in this proportion of patients on top of preventing future gout flares. Despite participants receiving ULT being more likely to be overweight or obese when compared to those not receiving ULT, lower lipid levels among those receiving ULT were seen. Additionally, lower CRP levels were observed in those receiving ULT compared to those not receiving ULT. Gout patients receiving ULT may garner added health benefits beyond lower urate levels. Future prospective longitudinal studies should further evaluate the clinical implications of chronically elevated serum uric acid and the impact of gout treatment on the incidence and management of gout comorbidities.

**Author Contributions:** Conceptualization, Y.M.R.; methodology, Y.M.R. and A.P.; formal analysis, M.O.-U., J.B.-G. and A.P.; data curation M.O.-U., J.B.-G. and A.P.; writing—original draft preparation, M.O.-U. and J.B.-G.; writing—review and editing, M.O.-U., J.B.-G., A.P. and Y.M.R.; supervision, A.P. and Y.M.R.; final approval, Y.M.R. All authors have read and agreed to the published version of the manuscript.

**Funding:** This research received no external funding.

**Institutional Review Board Statement:** All NHANES surveys are approved by the NCHS Research Ethics Review Board (https://www.cdc.gov/nchs/nhanes/irba98.htm, accessed on 3 November 2022) and all participants provide informed consent. Most NHANES data are made publicly available and are deidentified. Only publicly available data was used in this study and no institutional review board approval was required (45 CFR §46.102(f)).

**Informed Consent Statement:** Not applicable.

**Data Availability Statement:** NHANES data are publicly available.

**Conflicts of Interest:** Y.M.R. is an employee of the U.S. Food and Drug Administration. The views and opinions presented here represent those of the authors and should not be considered to represent advice or guidance on behalf of the U.S. Food and Drug Administration. The authors declare no conflict of interest.

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
