# Peer review of "Urate-Lowering Therapy Use among US Adults with Gout and the Relationship between Patients’ Gout Treatment Status and Associated Comorbidities"

_2674-0621, doi:10.3390/rheumato3010006_

Round 1

Reviewer 1 Report

In this article, Ortiz-Uriarte M et al. assessed the biomaker and  the prevalence of chronic comorbidities by gout treatment status. It is interesting to note that the prevalence of ULT use was 28.9% (very low) among patients in the United States with gout. However, the table are not clear and the discussion in the Discussion section seems inadequate. Therefore, we believe that there are serious problems with this manuscript that need to be addressed by the authors, as indicated below.

Major Comment

1. The authors described those on ULT had significantly lower LDL, HDL, and total cholesterol levels when compared to those not on ULT. This may be due to the fact that, as noted in Limitations, non-measurable parameters such as the use of antihyperlipidemic medications were not evaluated. Moreover, they described individuals on ULT were more likely to have a higher level of health insurance coverage. This could conceivably mean that those who are doing ULT are likely to be doing LLT as well. Therefore, the discussion section on dyslipidemia should be significantly changed.

2. The authors should add an entry for diabetes to Table 2.

3. The authors described the nephrotoxic effect of allopurinol is rare. However, because allopurinol is renally excreted, blood drug concentrations increase in CKD patients. This also increases adverse drug reactions, which is not uncommon.

4. The authors described allopurinol and febuxostat have been associated with improved treatment outcomes in heart failure patients and reduced risk of acute cardiovascular events in introduction section. I would suggest that this be stated a little more in the discussion section. It may be possible to use the literature on hyperuricemia patients without gout as well as hyperuricemia patients with gout.

5. Tables are difficult for readers to understand. Therefore, it is advisable to rewrite all of them for clarity.

Minor comments

1. Several typographical errors in the references are found.

Author Response

Reviewer 1

In this article, Ortiz-Uriarte M et al. assessed the biomaker and  the prevalence of chronic comorbidities by gout treatment status. It is interesting to note that the prevalence of ULT use was 28.9% (very low) among patients in the United States with gout. However, the table are not clear and the discussion in the Discussion section seems inadequate. Therefore, we believe that there are serious problems with this manuscript that need to be addressed by the authors, as indicated below.

Response: Thank you for taking the time to critically review our manuscript. Your questions and suggestions were helpful to us and significantly improved the quality of our manuscript.

Major Comment

  1. The authors described those on ULT had significantly lower LDL, HDL, and total cholesterol levels when compared to those not on ULT. This may be due to the fact that, as noted in Limitations, non-measurable parameters such as the use of antihyperlipidemic medications were not evaluated. Moreover, they described individuals on ULT were more likely to have a higher level of health insurance coverage. This could conceivably mean that those who are doing ULT are likely to be doing LLT as well. Therefore, the discussion section on dyslipidemia should be significantly changed.

Response: Thank you for your comment and suggestion. We agree with the reviewer regarding the effect of ULT on lipid parameters. But when adjusting for variables that influence receiving lipid-lowering drugs, the ULT was still significantly associated with lower LDL and TC. Though the results were surprising to us, some emerging evidence suggest that chronic ULT use can suppress the expression lipid-related genes and hence a favorable lipid profile. Furthermore, when adjusting for covariates of age, sex, education, race/ethnicity, BMI, and diabetes, hs-CRP levels were significantly lower among ULT users. We also acknowledge that higher socioeconomic factors can favor the results toward a better lipid profile in the ULT group. However, we adjusted the association to the same factors and the results remained the same. Text to address these major comments, we significantly revised and expanded the discussion. Also, we highlighted the issue of medications in the limitations section.  We address the reviewer's concerns by summarizing a study that concluded that there is an association between gout and dyslipidemia.

  1. The authors should add an entry for diabetes to Table 2.

Response: Thanks for the comment. A row summarizing the requested data was added in Table 2.

  1. The authors described the nephrotoxic effect of allopurinol is rare. However, because allopurinol is renally excreted, blood drug concentrations increase in CKD patients. This also increases adverse drug reactions, which is not uncommon.

Response: Thanks for the comment. We agree with the reviewer to expand the discussion further. The text further discussing the effects of ULT on the kidneys was added in the second paragraph of the Discussion section (lines 248-259)

  1. The authors described allopurinol and febuxostat have been associated with improved treatment outcomes in heart failure patients and reduced risk of acute cardiovascular events in the introduction section. I would suggest that this be stated a little more in the discussion section. It may be possible to use the literature on hyperuricemia patients without gout as well as hyperuricemia patients with gout.

Response:  Thanks for the comment and suggestion. The text further discussing the effects of ULT in the kidneys was added in the third paragraph of the Discussion section- last three sentences (lines 277-289).

  1. Tables are difficult for readers to understand. Therefore, it is advisable to rewrite all of them for clarity.

Response: Thanks for your comments and suggestions. We significantly changed all the tables (1-3) to improve readability.

Minor comments

  1. Several typographical errors in the references are found.

Response: Thanks for the comments. We reviewed and revised the references. The references were reviewed for typographical errors and were fixed accordingly.

Reviewer 2 Report

Urate-Lowering Therapy Use Among US Adults with Gout and the Relationship Between Gout Treatment Status and Associated Comorbidities

In this manuscript, Uriarte et al. have explored the role of comorbidities associated with gout and the potential benefits of urate-lowering therapies in relation to chronic illnesses such as diabetes, kidney, and heart diseases . Further exploration to understand the association between gout treatment status and comorbidity control would provide valuable insight into the optimal management of gout. This is an interesting study; however, it can be improved.

  1. The authors need to mention the statistical test used in the table. Table 1. Sample characteristics of U.S. adults 30 years or older being told by a physician they have gout.
  2. The authors should explain their sample selection criteria and the methods used to calculate the sample size, effect size, and power of the study. They should also consider potential sources of bias and provide an analysis of how this bias may affect their results.
  3. the authors should expand on the methodology used to calculate Odds ratio.

“Table 2. Odds ratios for the prevalence of comorbid conditions among those on urate-lowering therapy among adults 30 168 year or older having been told by a doctor they have gout.”

  1. The authors need to perform multivariate regression to identify the variables. The regression results would be better reflective of the significant clinicopathological factors.

Author Response

Reviewer 2

Urate-Lowering Therapy Use Among US Adults with Gout and the Relationship Between Gout Treatment Status and Associated Comorbidities

In this manuscript, Uriarte et al. have explored the role of comorbidities associated with gout and the potential benefits of urate-lowering therapies in relation to chronic illnesses such as diabetes, kidney, and heart diseases. Further exploration to understand the association between gout treatment status and comorbidity control would provide valuable insight into the optimal management of gout. This is an interesting study; however, it can be improved.

Response: Thank you for taking the time to critically review our manuscript. Your questions and suggestions were helpful to us and significantly improved the quality of our manuscript.

  1. The authors need to mention the statistical test used in the table. Table 1. Sample characteristics of U.S. adults 30 years or older being told by a physician they have gout.

Response: Thank you for the comment and suggestion. Details of the statistical tests used in each table are now detailed in the respective legends.

  1. The authors should explain their sample selection criteria and the methods used to calculate the sample size, effect size, and power of the study. They should also consider potential sources of bias and provide an analysis of how this bias may affect their results.

Response: Thanks for the comments. To address this comment, we created a flowchart (Figure 1) showing how the study sample population was created.

  1. the authors should expand on the methodology used to calculate Odds ratio.

“Table 2. Odds ratios for the prevalence of comorbid conditions among those on urate-lowering therapy among adults 30 168 year or older having been told by a doctor they have gout.”

Response: Thanks for the comment. Text was added in the methods section describing the interpretation of the calculated odds ratios. Please see lines 150-152

  1. The authors need to perform multivariate regression to identify the variables. The regression results would be better reflective of the significant clinicopathological factors.

Response: Thank you for the comment and suggestion. We conducted a multivariable linear regression analysis. Please see the method section and tables 2-3 and lines 141-156.

Round 2

Reviewer 1 Report

The authors sincerely responded to the reviewers' comments and revised the manuscript. We therefore consider this manuscript to be in good condition for submission.

Reviewer 2 Report

The authors have improved the manuscript.